# Sulfur in Seeds: An Overview

**DOI:** 10.3390/plants11030450

**Published:** 2022-02-06

**Authors:** Sananda Mondal, Kalipada Pramanik, Debasish Panda, Debjani Dutta, Snehashis Karmakar, Bandana Bose

**Affiliations:** 1Department of Crop Physiology, Institute of Agriculture, Visva-Bharati University, Sriniketan 731236, India; debasish.panda@visva-bharati.ac.in; 2Department of Agronomy, Institute of Agriculture, Visva-Bharati University, Sriniketan 731236, India; kalipada.pramanik@visva-bharati.ac.in; 3Department of Plant Physiology, Faculty of Agriculture, Bidhan Chandra Krishi Viswavidyalaya, Mohanpur 741252, India; dutta.debjani@bckv.edu.in (D.D.); karmakar.snehashis@bckv.edu.in (S.K.); 4Department of Plant Physiology, Institute of Agricultural Sciences, Banaras Hindu University, Varanasi 221005, India; bbosebhu@gmail.com

**Keywords:** sulfur, sulfate transporters, seed storage proteins, seed germination, seed priming

## Abstract

Sulfur is a growth-limiting and secondary macronutrient as well as an indispensable component for several cellular components of crop plants. Over the years various scientists have conducted several experiments on sulfur metabolism based on different aspects of plants. Sulfur metabolism in seeds has immense importance in terms of the different sulfur-containing seed storage proteins, the significance of transporters in seeds, the role of sulfur during the time of seed germination, etc. The present review article is based on an overview of sulfur metabolism in seeds, in respect to source to sink relationships, S transporters present in the seeds, S-regulated seed storage proteins and the importance of sulfur at the time of seed germination. Sulfur is an essential component and a decidable factor for seed yield and the quality of seeds in terms of oil content in oilseeds, storage of qualitative proteins in legumes and has a significant role in carbohydrate metabolism in cereals. In conclusion, a few future perspectives towards a more comprehensive knowledge on S metabolism/mechanism during seed development, storage and germination have also been stated.

## 1. Introduction

Sulfur is an essential macronutrient; according to its chemical nature it plays important roles in various cellular metabolic processes [1]. As an organic thiol it is incorporated in essential amino acids such as cysteine, methionine and different co-enzymes (biotin, coenzyme A, iron-sulfur proteins, ethylene, thiamine pyrophosphate and lipoic acid), and thioredoxins and sulfolipids are often responsible for the structure and biological activity of proteins [2,3]. Recently, it has been noted that a deficiency of sulfur in soil is a problematic issue which leads to reduction in yield and quality of seeds. Day by day the concentration of sulfur in soil is decreasing due to the increasing proportions of high analysis and sulfur-free fertilizers and the decreasing use of traditional organic manures in agricultural land. Sulfur is taken up by a plant’s roots in the form of sulfate from the soil, which is also a transportable form in the plant system [4]. Moreover, Abou Seeda et al. [2] revealed that the primary uptake mechanism of sulfate takes place via roots, and afterwards, it is translocated to various locations within the plant body through xylem. Assimilation of sulfur is occurring in the form of cysteine; it acts as a precursor/donor of reduced sulfur mostly for other several organic sulfur compounds present in plants [5]. In rice plants leaf, a significant increment in carbohydrates (starch) was noted when the plants were grown in sulfur deficient soil [6].

It has been observed that, without presence of adequate sulfur (S) in soil, plants/crops cannot complete their life cycle adequately in terms of yield, quality or protein content in seeds, nor can they make optimum utilization of supplied N, required for protein and enzyme synthesis [7]. Sulfur compounds like tripeptide glutathione (GSH) are involved in detoxification of reactive oxygen species (ROS) in response to various abiotic and biotic stresses; it also acts as a substrate for the biosynthesis of phytochelatins and/or glucosinolates (GSLs) in response to defense against herbivores and other pathogens [8]. Similarly, Honse et al. [9] reported that sulfur deficiency leads to reduction in sulfate as well as S-containing compounds like cysteine (Cys), methionine (Met) and GSH; protein synthesis was suppressed and the pools of soluble N, such as nitrate and amides, were also increased. During the time of seed maturation, the GSH/GSSG (glutathione disulfide) ratio was increased, whereas glutathione reductase (GR) activity decreased. Moreover, the glutathione level became very low while imbibing, resembling ROS production during radicle protrusion. At the time of maturation, GSH accumulation signifies a protective role of GSH as an antioxidant during the storage of seeds. Along with this, the storage pool of S is remobilized, and the amount of S-rich seed storage proteins (SSP) is decreased, whereas to compensate for the deduction of total seed proteins, the S-poor SSP is enhanced [10].

Crop yield and seed quality assessment are closely related with the sulfur content of the seed [11,12,13]. In different plants, the S-containing amino acids are an important determining and evaluating index for seed quality assessment. As mentioned previously, the S-limited condition reduces the proportions of S-containing amino acids in grains, whereas its application increases the content also. In addition to this, it increases the nutritional value of the grains as well [13,14]. Furthermore, various studies with different crops emphasized that the supply of S in varying levels affects the extraction rates of wheat flour, quality of gluten and baking properties [15]. In the case of forage grass, it also affects the seed yield and quality of seed, and by applying them with an optimum quantity, these traits can be improved, as well as the nutritional status of herbivorous livestock, and as a result, it increases the yield and quality of wool, produces more milk and improves the quality of milk of dairy cows [16]. In addition, S also affects the N fixation in legume crops. As per the report of Scherer et al. [17] and Schneider et al. [18], the application of S fertilizers could have the capacity to enhance the N fixation in pea (*Pisum sativum* L.) and alfalfa (*Medicago sativa* L.).

Tabe and Droux [19] conducted an experiment by applying sulfur in different crop plants and it was found that sulfur is delivered to seeds through the phloem as sulfate in pods of legumes, or in reduced forms such as GSH in rice grains and *S*-methylmethionine in wheat. It was again revealed by different scientists that the sulfur-reducing enzymes are present in developing seeds and the reduced sulfur accumulated in mature seeds of grain legumes [19,20,21] and wheat [22]. In contrast, rice seeds received an amino form of S as nutrition [23]. Translocation of sulfate by phloem-localised sulfate transporters contributes significantly to importing S in seeds, as reported by Awazuhara et al. [24]. In addition, Tabe and Droux [21] depicted that in grain legumes developing cotyledons are capable of assimilating sulfate in the form of sulfur amino acids, which are presumably acquired from the seed apoplasm and released from the seed coats. Interestingly, a voltage-dependent SO_4_^2^^−^-permeable channel was identified in plasma membranes of protoplasts, derived from seed coats of developing chickpea (*C. arietinum* L.) seeds [25]. Furthermore, Tabe and Droux [26] observed that in seed coats of chickpea (*Cicer arietinum* L.) a putative sulfate transporter gene (*Sultr3-1*) is expressed, but not expressed in developing embryos.

The metabolism of seeds in terms of carbohydrates, proteins and fats/oils is regulated by sulfur, similarly to other nutrients, following the criteria of essentiality. Soybean seeds contain up to 0.46% (*w*/*w*) of sulfur. In general, leguminous seeds reserve a lower amount of carbohydrates as compared to cereals. Legume seeds consist of 35 to 45% protein, which makes the diet proteinaceous, fibrous and associated with some therapeutic uses for diseases such as diabetes and cancer. Legume seeds contain different proteins such as globulins, which act as a major source of protein. Albumins are recognized as reserve proteins in legume seeds [27]. In this context, deficiency and/or toxicity of sulfur in the form of sulfate affects the growth and development of plants as well as seed metabolism of soybean (Glycine max (L.) Merr.). Its optimal supply is essential to run the normal life cycle of plants and also to maintain the quality aspect of seeds. In soybean the nutritive value and the storage reserve of seeds are regulated by sulfur, which is an essential factor to maintain the seed yield and seed quality [27].

Regarding the facts associated with oilseeds and pulses in respect to the essentiality of S, especially in canola (*Brassica napus* L.) and soybean (*G. max* L.) growth periods, it was observed that for the synthesis of oils and proteins in pulses and oil seeds S is required [28]. In oilseed rape, the application of S fertilizers improves N-use efficiency and maintains a sufficient oil quantity and quality of fatty acids profile [29,30]. D’Hooghe et al. [31] did an experiment on oilseed rape (*B. napus* L.), which was based on a proteomics platform of mature seeds obtained from winter oilseed rape plants grown under low sulfate applied at the bolting, early flowering or initiation of pod filling stages. It was noted that the seed protein quality was poor based on the severity of S limitation and was associated with a deduction in S-rich SSP accumulation (Cruciferin Cru4), which favored S-poor SSP (Cruciferin BnC1).

The leaves of oilseed rape (*B. napus* L.) recycle the foliar compounds and contribute to seed filling during the time of the reproductive stages, which may lead to better seed yield with good quality oil. If S limitation occurred at the rosette stage in winter oilseed rape, they maintained their growth via recycling of endogenous foliar S compounds, specifically sulfate, in optimum quantity from old and mature leaves to the new ones without accelerating any leaf senescence [32,33]. Although S mobilization from vegetative tissues is important during the time of seed filling, little information is available about the dynamics, the efficiency and the contribution of S mobilization from vegetative tissues to oilseed rape seeds.

Based on the scientific reports/literature, it is clear that sulfur (S) is a growth limiting macronutrient of plants, required in different growth phases of plants. From an agricultural point of view, seed yield and quality of seed are important traits which are regulated by S, whether it is cereals, pulses, oilseeds and in many other crops. The aim of the present review article is to include the different aspects of seeds regulated by sulfur, i.e., source to sink relationships, S transporters present in the seeds, S-regulated seed storage proteins present/accumulated in the seeds and the role of sulfur in seed germination.

## 2. Source to Sink Relationship

Sulfur is taken up from the rhizospheric region by plant roots as sulfate and distributed within the tissues in this form. Moreover, Tabe and Droux [21] depicted that the dominant form of sulfur is sulfate and it is translocated via phloem, supplied in pods during the time of lupin (*Lupinus albus* L.) seed development, then the seed is able to reduce and assimilate sulfate. Within the plant tissue, including the developing phase of seeds, if sulfate is not in a reduced form, then the excess amount of sulfate is stored in the vacuole [34,35]. In developing seeds of *M. truncatula,* a transcriptomic analysis revealed that within the seed tissues sulfur assimilation takes place by following two distinct pathways (Figure 1). In most of the cases, sulfate enters the embryo and is utilized for the biosynthesis of cysteine (Cys) and gets incorporated into proteins, while in another pathway sulfate enters into the seed coat and endosperm. The latter is preferentially involved in the biosynthesis of defense-related sulfur compounds [36]. This kind of partitioning of sulfur in between the seed compartment implies an active exchange of sulfate.

## 3. Sulfate Transporters: Regulate the Sulfur Translocation in Seeds

The beginning of the sulfate uptake mechanism by the root tissue from the surrounding environment and the translocation of sulfate between different cell compartments is facilitated by specific sulfate transporters (SULTR) (Figure 1). The expression levels of these transporter genes in specific organs, cell types and subcellular compartments are regulated by the transcriptional and post transcriptional mechanism which maintains the homeostatic balance between the uptake of sulfate and internal tissue distribution on the basis of sulfate availability and on-demand organic sulfur metabolites biosynthesis [37]. SULTR, a large gene family, is involved in encoding this transporter which consists of 14 members in Arabidopsis and rice (*Oryza sativa L.*). According to phylogenetic studies this gene family can be categorized into 4 closely related groups (*SULTR1* to *4*), each containing 12 membrane-spanning domains and a STAS (sulfate transporter and anti-sigma antagonist) domain at the carboxy-terminal end [38], and a fifth member of this group is SULTR5, distinct from the other, lacking that STAS domain [39]. Interestingly, Tomatsu et al. [40] mentioned that the Arabidopsis *Sultr5;2* gene was responsible for encoding a high-affinity root molybdate transporter, which raises a valid question about the role of group 5 transporter genes in the sulfate transport mechanism. Furthermore, group 1 and 2 transporters of sulfate are localized in the plasma membrane and are considered as the best categorized groups, being subjected to several studies. Members of group 1 sulfate transporters represent a high-affinity transport system which facilitate the uptake of sulfate by roots (SULTR1;1 and SULTR1;2) or translocate the sulfate from source to sink tissues (SULTR1;3) [41,42,43,44,45]. In addition, the group 2 members consist of low-affinity sulfate transporters whose gene products may be involved in vascular tissues transportation and facilitate the translocation of sulfate throughout the plant [24,39,44]. The literature suggests that group 3 is composed of low-affinity transporters localized at the plasma membrane, showing differential expression patterns in plant tissues and not stimulated by sulphur deficiency [39]. In addition, a role of the SULTR3;5 transporter is noted in the transport of sulfate from root-to-shoot in cooperation with the SULTR2;1 transporter of Arabidopsis [46]. The last group, i.e., group 4 sulfate transporters, has been identified in the vacuolar membrane: a study with SULTR4;1-GFP fusion protein showed that this transporter mainly accumulated in the vacuoles of roots and hypocotyls of young seedlings [46]. Under sulphur sufficient and deficient conditions, the *Sultr4;1* transporter gene was expressed in roots and helps to efflux sulfate (SO_4_^2-^) from the vacuolar lumen to cytoplasm, and it also improved the storage capacity of sulfate in vacuoles [46]. In contrast, gene expression of *Sultr4;2* was highly inducible by sulfur during the time of sulfur deficient condition in the same tissue. The double knock-out mutants, i.e., *sultr4;1/sultr4;2*, contained higher amounts of sulfate as compared to wild-type plants. Comparison between the single and double knock-out mutants *sultr4;1*/*sultr4;2* demonstrated that *Sultr4;1* devotes a vital role and *Sultr4;2* has a supplementary effect [47]. Whereas the sulfate transport system has been studied extensively in roots, to date, there are very few reports available based on the functions of individual sulfate transporters within seeds. Moreover, vacuoles may play an important role for the purpose of storage and unloading of sulfate within the developing seed, where the members of the SULTR4 transporters would play a key role. In this context, the Arabidopsis *Sultr4;1* gene was expressed strongly within the developing seeds, and it was observed that its disruption significantly enhanced the content of seed sulfate, depicting that SULTR4;1 was involved in the efflux mechanism of sulfate from vacuoles to the developing seeds. In addition, a proteomic study of *Sultr4;1* mutant seeds revealed the metabolic adjustment for the adaptations in altered sulfate compartmentalization, which indicates a SULTR4;1-mediated sulfate transport system for the establishment of defense mechanisms against oxidative stress during the time of seed development.

For instance, in mature mutant seeds of *sultr4;1*, on average, the sulfate content was 1.7 times more than that of wild-type plants, whereas the total sulfur content in seeds remained unchanged. In mature seeds, sulfate contributes a significant fraction of the total sulfur content, i.e., 7.7% in wild-type and 13.24% in *sultr4;1* mutant seeds, respectively. In respect to mutant *sultr4;1* plant phenotypic response, no significant difference was observed in terms of yield parameters, leaf area and/or onset of flowering but a slight reduction in seed weight was noted as compared to wild-type plants. The sulfur ion fluxes into the developing seeds may not be affected in the *sultr4;1* mutant, which increases the sulfate contents in seed, but there is no relation for such a drastic perturbation of vegetative growth of this mutant. In mature seeds of the *sultr4;1* mutant, a significant increment of sulfate pool was observed, which may be related to a reduced efflux of sulfate from the vacuoles during the development of seeds. In continuation with this, in the reproductive growth phase of Arabidopsis, the *Sultr4;1* transporter was preferentially expressed in developing seeds during the time of transition between embryogenesis and the seed filling phase and a relatively higher amount of transcript availability was observed in comparison to *Sultr4;2*, which was expressed at equal levels throughout the development of seeds. Additionally, the transporter SULTR4;1 also maintains the redox homeostatic balance at the time of seed development if any kind of oxidative stress outbreak takes place due to any environmental abnormalities. The proteomic study revealed this kind of dehydration tolerance capability with the due course of their development [48,49]. In a nutshell, it can be suggested that the sulfate transporters SULTR3 and SULTR4 play a vital role in sulfate translocation, which is associated with seed development, by suppling sulfate and different S metabolites. In this context, it can also be noted that SULTR3 and SULTR4 homologs control the allocation of sulfate in between the seed compartments and help to modulate S metabolites and seed protein composition in Arabidopsis [48,49].

According to Zuber et al. [50], it was noted that different members of the SULTR3 family were more highly expressed at different stages of seed development of Arabidopsis than the other organs, which have the ability to control the translocation of sulfate within developing seeds. For instance, in developing embryos of chickpea (*C. arietinum* L.), various homologous members of the SULTR3 sub-type transporters were involved in sulfate transport and delivery as reported by Tabe et al. [51]. 

Another experiment depicted that, at bolting stage, previously stored sulfate in vacuole of the source leaves is remobilized into the developing seeds of oilseed rape (*B. napus* L.) and an up-regulated gene expression is noted of *SULTR4;1* and *SULTR4;2* transporters. These kinds of upregulated gene expression of two SULTR4-type transporters were also observed during the vegetative growth phase in old and mature leaves [33] and in roots [52] of oil seed rape (*Brassica napus* L.). In addition, Gironde et al. [53] revealed that at the reproductive stage, sulfate is the main source of S remobilized from the stored vacuolar sulfate of leaves via tonoplastic SULTR4-type transporters in S deficient condition. Moreover, Awazuhara et al. [24] revealed that sulfate transporter SULTR2.1 is also involved in the transfer of S into developing seeds of Arabidopsis. In addition, when the amount of abscisic acid (ABA) in freshly harvested seeds of *sultr3;1* mutants was tested, 25–50% more ABA was found in comparison to wild-type Arabidopsis plants; this finding established the fact that SULTR3;1 affects ABA biosynthesis not only during the time of early vegetative growth but also in the seed filling stage [54].

## 4. Sulphur Containing Seed Storage Proteins (SSPs)

Seed storage proteins (SSPs) are the essential resources of nitrogen, carbon, and sulfur required for the germination of seeds, and their amount varies according to the availability of nutrients in the rhizosphere [55]. Two major types of storage proteins were found in Arabidopsis seeds, i.e., 12S globulins or cruciferins (saline-soluble) and 2S albumins or arabidins (water-soluble) [56]. In response to sulfur deficiency (S), the SSPs present in different plants showed a similar pattern of behavioural responses. Similarly, in sulfur limited condition, the amount of sulfur-rich proteins such as 12S globulins and 2S albumins in Arabidopsis or 11S globulins (glycinin) in soybean (*G. max* L.) are decreased [55,57], whereas the sulfur-poor SSPs, such as β-conglycinin (the 7S globulin), in developing seeds of soybean were accumulated [57]. Moreover, to maintain the normal growth and development, plants accumulate nitrogen as a source of seed protein, even under sulfur deficient conditions [55]. Furthermore, the application of the immediate precursor of cysteine biosynthesis, i.e., *O*-acetylserine (OAS), in immature cotyledons of soybean showed a similar kind of SSP accumulation to that observed under sulfur deficiency [58,59]. Therefore, OAS can be considered as a regulator of SSP gene expression [58]. Furthermore, sulfur deficiency induced (*SDI*) genes *SDI1* and *SDI2* have been demarked as OAS responsive genes [60,61] and have the ability to downregulate the S-rich secondary metabolites, i.e., GSLs, in shoots as well as roots of Arabidopsis via interacting with MYB28 (MYB; myeloblastosis) proteins present in the nucleus [62]. For instance, Gao et al. [63] revealed that *SDI1* is highly expressed under sulfur deficiency (S) in seeds, which has an additional role for the modulation of the SSP profile in favour of S-poor proteins by interacting with MYC2 (MYC; master regulator of cell cycle entry and proliferative metabolism) transcription factors (TFs), and participated here, and MYB28 was also introduced in this process. Whereas, under sulfur limited condition, *SDI1* is expressed in seeds and during the time of the sulfur assimilation pathway and metabolism, it co-ordinately downregulates these two main sulfur-rich pools, i.e., GSLs and S-rich SSPs (Figure 1). Metabolomic analysis demonstrated a distinct metabolic change during the time of seed development upon *SDI* perturbation; it includes changes in amino acids, organic acids and sugars, and it mimics the responses observed under S deficient condition; this observation was further validated by specifying the role of *SDI* under S deprivation [64,65]. This study partly explains the crucial role behind the high expression of *SDI* genes during later stages of seed maturation. However, a regulatory mechanism should exist in seeds to finetune the *SDI* expression over the period of seed development. The expression of *SDI* is repressed by an unknown mechanism under favourable nutritional conditions during the early to mid-phase of seed maturation, whereas at the late to post maturation phase, under S deficient condition or when sulfur pools have been already used up for S-rich SSPs synthesis, *SDI* gene expression is activated, to maintain the balance between S-rich to S-poor proteins and perhaps end the cellularization period. Moreover, a common response of S deficiency condition is an enhancement of the root-to-shoot ratio, where shoot growth is more reduced than the root growth [66,67,68,69].

The sulfur deficiency-induced proteins SDI1 and SDI2 have an important role under sulfate-deprived conditions (S), i.e., to maintain sulfur homeostasis by downregulating glucosinolates. SDI1 also downregulates another sulfur pool, which was the S-rich 2S SSPs synthesis in Arabidopsis (*Arabidopsis thaliana*) seeds. It was also noted that MYB28 directly regulates the 2S SSPs by binding to the At2S4 promoter. In this extension, *SDI1* directly downregulates the 2S SSPs by forming a ternary protein complex with MYB28 and MYC2 (another transcription factor regulates the SSPs) [70]. These findings will help a lot in the future to understand plant responses towards sulfur deficiency.

Neequaye et al. [71] did a field trial with field grown *B. oleracea* using CRISPR-Cas9-mediated gene editing technology to signify the role of MYB28 in the regulation of aliphatic glucosinolate biosynthesis and the associated sulphur metabolism in leaves and floret tissue. They reported that knock-out mutants of *myb28* result in downregulation of aliphatic glucosinolate biosynthesis genes and reduction in accumulation of methionine-derived glucosinolate, glucoraphanin, in leaves and florets tissue of field-grown *myb28* mutant *B. oleracea* plants. This experiment also indicates the significant importance of the transcription factor MYB28 in sulfur metabolism in all developmental stages of B. oleracea, maybe in seed development and germination also. Future studies will reveal this gap. 

In the senescence phase of a plant’s life, amino acids are remobilized and transported from the old mature tissue to the seeds and energy is recovered from reduced N and S for the next generation. As seeds contain low levels of free amino acids during desiccation, they have to store them in the form of proteins compatible with seed dormancy and easily hydrolysable at the time of seed germination. In addition, Cys and Met have to be considered in a different way as these are the only two amino acids out of twenty which contain sulfur. Interestingly, maize (*Zea mays* L.) embryos contain on an average more lysine in comparison to Met, whereas the endosperm has higher levels of Met than Lys [72,73].

Unlike accumulation of nitrogen in seeds, sulfur storage is regulated via accumulation of a few specialized sulfur-containing proteins in maize endosperm. In maize seeds (Z. mays L.), β- and δ-zeins are the major sink for Met and β- and γ-zeins for Cys. The reason behind this is Met constitutes the endpoint of the sulfur amino acid biosynthetic pathway and the sulfur moiety attaches with the Cys, which enhances the expression pattern of Met-rich proteins, preventing the accumulation of Cys-rich proteins at endosperm development. It has been suggested that the increased content of seed Met may be the cause for the reduction of sulfur in the leaves during the time of photosynthesis [73]. Literature suggested that S-rich proteins should be introduced within the developing seeds, at the same time as a reduction in the endogenous level of S-rich proteins is observed, where a reallocation of seeds’ S protein was recommended under limitations of S availability [23,74,75]. In this context, Planta et al. [76] produced an engineered sulfur storage transgenic maize (*Z. mays*) kernel having the capacity to express higher levels of Met-rich 10 kDa δ-zein and total sulfur protein without reducing the other zein proteins or causing an apparent yield loss. 

Delivery of adequate sulfur into the seed tissues is required for maximization of the production and to improve the quality of seed proteins [77]. The sulfate uptake and assimilation are based upon the nutrient status of the plant [78]. Several studies showed that decrease in sulfate availability enhanced the expression of sulfate transporter genes several-fold, and it increased the sulfate uptake capacity [79]. Application of sulfur fertilizer preferentially affects the quality of proteins by increasing the expression of sulfur-rich amino acid-containing proteins. According to the sulfur status the seed protein accumulation was regulated and has been well documented in many legumes, including globulins in soybean (*G. max* L.) and lupine (*Lupinus* sp.), as well as globulins and albumins in pea (*P. sativum* L.) [78].

In addition, *Phaseolus vulgaris* and several *Vigna* sp. are able to accumulate high levels of nonprotein amino acid such as *S*-methyl-Cys in the form of a *γ*-Glu dipeptide in seeds [80]. Moreover, the results showed that combined deficiencies in phaseolin, phytohemagglutinin and arcelin in common bean seeds lead to a significant rise in sulfur amino acid content, especially Cys, in the form of *S*-methyl-Cys and *γ*-Glu-*S*-methyl-Cys. Based on natural genetic variation in the composition of sulfur containing SSP, a strategic improvement of the nutritional aspects of common bean seeds is possible, as reported by Taylor et al. [81].

We can consider an alternative transgenic approach where gene manipulation may be a way out for the improvement of sulfur amino acid pathways. A 70% increment in total Cys concentration was observed in developing soybean (*G. max* L.) seeds due to overexpression of cytosolic serine acetyltransferase [82]. Song et al. [83] revealed that feedback-insensitive Arabidopsis cystathionine γ-synthase (*AtD-CGS*) gene expression, encodes a protein, lacks 30 amino acids units from the N-terminal end, and as a result, the total Met concentration was raised up to 1.8- to 2.3-fold, which was an overall enhancement in seed proteins. In contrast with the earlier experiment, feedback-insensitive *mto1-1* allele’s expression harbors a point mutation which causes an elevated level of free Met, but not the amount of total Met in soybean (*G. max* L.) seeds, however, in adzuki bean (*V. ungularis* L.), the cystathionine levels were raised where the total concentration of Met was originally decreased [84]. Similarly, Pandurangan et al. [78] depicted that optimum sulfur levels in plant tissue as a nutrition is essential to maximize the concentration of amino acids and this improves the quality of S-containing SSPs in genotypes (SMARC1N-PN1) lacking the phaseolin and major lectins in common beans (*P. vulgaris* L.).

Under mild sulfur deficient condition, the sulfur-containing proteins levels like legumin and pea albumin 1 (PA1) are reduced in the seeds of pea (*P. sativum* L.), whereas in the same condition the poor sulfur-containing proteins like lectin and vicilin are either unaffected or increased slightly. This study indicates the level of different sulfur-containing proteins which is a reflection of respective mRNAs of the proteins. Interestingly, vicilin and convicilin are a major potential allergen isolated from pea seeds. Furthermore, vicilin’s proteolytic fragments are depicted as relevant IgE binding components in pea [85]. In another experiment, the involvement of amyloid formation during the time of accumulation of storage proteins in plant seeds was revealed for the first time. They observed that garden pea (*P. sativum* L.) seeds contain amyloid-like aggregates in storage proteins, the predominant one being 7S globulin vicilin, which form bona fide amyloids in in vivo and in vitro condition. Vicilin proteins consists of two evolutionary conserved β-barrel domains. The accumulation of vicilin amyloid enhanced at the time of seed maturation and germination. Amyloids of vicilin resist digestion by gastrointestinal enzymes and exhibit toxicity in yeast and mammalian cells [86]. 

## 5. Sulfur Regulated Seed Germination

Germination is the most important and the beginning phase of any plant’s life cycle. Seed germination is divided into three phases, during this course of time various metabolic activities take place including sulfate metabolism. Very little information is available regarding seed germination influenced by S-containing SSP and various S-regulated functional proteins and stored phytohormones present in seeds. Based on sulfate metabolism, seed germination was assessed by creating multigene defective mutants under abscisic acid (ABA) and salt conditions. In the same situation, a single mutation in *SULTR3* transporters gene construct can alter seed germination responses. A progressive reduction in germination percentage was also observed from single to *SULTR3* quintuple mutants, the latter showing the lowest rate of germination under both ABA and NaCl treatment [87]. After 5 days of sowing, the *SULTR3* quintuple mutant showed only a 10% seed germination rate, whereas in the wild type, more than 60% germination was recorded under 0.3 mM ABA. Furthermore, at 8 days after sowing (DAS), only 20% of the *SULTR3* quintuple mutant seeds were germinated, whereas in the wild type, more than 90% of the seeds were germinated. A similar pattern was observed under salt stress [87]. Consistent with alterations in Cys and ABA content, seed germination inhibition upon treatment with exogenous ABA and salt was directly proportional to the number of mutated *SULTR3* genes, demonstrating the correlation of Cys and ABA content with the plant abiotic stress response during germination stages (Figure 1). In *SULTR3* quadruple and quintuple mutants, delayed germination of seeds was noted under normal condition, in comparison to the wild type. In the case of triple mutants, *sultr3;1 sultr3;2 sultr3;3*, but not *sultr3;2 sultr3;3 sultr3;4* or *sultr3;3 sultr3;4 sultr3;5*, showed a delayed germination at 3 DAS, which indicates that *SULTR3;1* may play a more crucial role than other *SULTR3* members during the early stage of seed germination of Arabidopsis [87].

Studies depicted that H_2_S is an essential component of sulfur and cysteine metabolism [86], is generated via different enzymatic and nonenzymatic mechanisms and acts as a signaling molecule in plant cells [88,89]. The metabolic process of H_2_S is dependent on the type of subcellular compartment and plant organ under optimal environmental conditions, as well as in stressful situation [88]. Seed physiological quality represented a higher rate of seed germination and longevity, lower reserves deterioration and higher seedling vigor. A reduced physiological quality of seeds decreases the germination percentage and vigor index of the seed [90], whereas the exposure of seeds in H_2_S under stress as well as normal conditions has generated positive effects on germination percentage and time, seedling growth, fresh weight, root length and other metabolic changes [91]. Moreover, Caverzan et al. [92] narrate that future studies are essential to understand the interaction between metabolic pathways related to H_2_S and how this signaling molecule is involved in improvement of seed germination process.

Germination is a very crucial process; here, a seed starts its second generation by coping with its new ambient conditions, which sometimes may create constrains for it. However, seed germination can be sped up via utilizing the knowledge of a new exciting technology, named seed priming, which is easy to handle, farmer friendly and helpful in sustaining nature, which led plant scientists to unveil a number of facts about the seeds via various omics-related studies and finally a new term also coined as “primeomics” [93]. However, Bose and Mishra [94] observed that only 24 h of pre-sowing soaking (one kind of seed priming technique) of mustard (*B. rapa*; a sulfur enriched crop) seeds with Mg(NO_3_)_2_ and MgSO_4_ salts improved the percent germination; furthermore, their finding validated that the affectivity can be carried over from vegetative phase to yield; this experiment depicted that NO_3_^-^ and/or SO_4_^2-^ might be a responsible factor for the improvement of yield. This may give support to the presence of low affinity sulfate transporters in seeds which may have worked effectively in the presence of MgSO_4_ solution, during the process of priming of mustard seeds [95,96].

## 6. Conclusions

The present review article is based on sulfur metabolism/content in seeds. As per the literature, leaves are the major source for the transportation of sulfur during the reproductive phase/seed development phases of plants. From an agricultural point of view, seed yield and the quality of a crop’s seed is a matter of seriousness, where S as a macronutrient has a great impact to improve these traits. Various scientists designed strategic experiments such as to induce more proliferation of roots, remobilization of sulfur from mature parts of plants and a sulfur transient storage pool in leaves and/or stem, which may help during the time of seed development and a strong source to sink relationship definitely has to be established. Very limited information has been extracted about the role of sulfate released from the seed vacuoles during the seed developmental phase to maintain cellular homeostasis [49]. However, a few members of the SULTR3 and -4 family transporters are noted to be involved in sulfur transportation mechanisms in the course of seed development, but the exact location and how they perform their work is still unclear. However, there is no information available regarding the involvement of specific transcription factors, mitogen-activated protein kinases (MAPKs), redox status, etc. in different cell types throughout seed development and storage. Further studies are required to understand the sulfur metabolism/mechanism during seed development and storage, which will help to improve the seed yield as well as the quality of seeds and ensure a good return to the farmers of developing nations and/or be of benefit to the whole world irrespectively.

## Figures and Tables

**Figure 1 plants-11-00450-f001:**
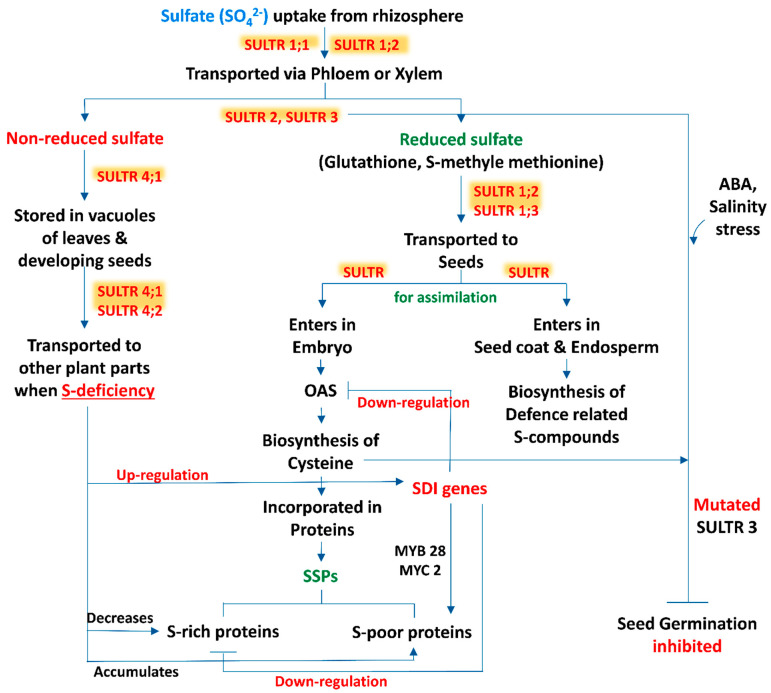
A schematic representation of source-sink relationship of sulfur (S) in plants: an insight in uptake and transportation of sulfate (SO_4_^2−^) in different plant parts by various transporters (SULTRs) and its assimilation, accumulation of S-containing seed storage proteins (SSPs) and regulation of S-mediated seed germination. (Abbreviations: OAS, *O*-acetylserine; SDI genes, sulfur deficiency induced genes; MYB28, myeloblastosis28; MYC2, master regulator of cell cycle entry and proliferative metabolism2; ABA, abscisic acid).

## Data Availability

Data sharing not applicable.

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
