# Peer review of "Sulfur in Seeds: An Overview"

_plants, 2022, doi:10.3390/plants11030450_

Round 1
Reviewer 1 Report
The review of the paper submitted to Plants by Mondal et al.
The review focuses on sulfur metabolism in seeds, both developing and germinating. The issue of sulfur metabolism is especially important for quality of seed production and cultivated plants yield. There is no review paper concerning this topic and it seems to be quite interesting to published it. The authors cited also newly published researches, however the papers published within the last 5 years constitute just for 18% of all cited papers. In review papers it should be more to keep the review up to date.
The paper need some English correction. F.eg. L47 abiotic and biotic stresses, L85 carbohydrates, proteins and fats
Arabidopsis should be written italics, as it is Latin name. The genes names also should be italics, as in several places are not (f.eg. L252, L280, L347 and some others places.
The function of MYB28 in sulfur metabolism was already confirmed by Mikhaela Neequaye et al. in field study on CRISP-Cas9-mediated gene edited B. oleracea (10.1101/2020.08.02.232819 and 10.1089/crispr.2021.0007).
I advised you to mention just in few sentences the engagement of sulfur in signaling thought H2S interaction. There are lots of new papers showing important function of H2S in signaling next to NO and H2O2 in plants, also in seed development and germination. In my opinion it is also an important aspect in sulfur metabolism.
Author Response
Sir, as per your recommendation I have corrected my manuscript and included a paragraph of this paper (10.1101/2020.08.02.232819 and 10.1089/crispr.2021.0007) as well as the the role of H2S in seed germination another paragraph. I hope it will make you statisfy.
Thank you Sir for your valuable recommendation.
Reviewer 2 Report
Dear Authors,
The manuscript by Mondal et al. “Sulfur in seeds: An overview”, reviews source-sink relationships, S transporters, S regulated seed storage proteins in the seeds, role of sulfur in seed germination. But this version of the MS needs to be improved. The following figure showing the work of the sulfur transporters in plants and especially in seeds helps with reading. I would also like to see a final diagram or figure that would summarize and help in reading the review. Without a final diagram or figure, reading the review is difficult and boring.
Minor comments
It is necessary to uniform the designations of genes: in some cases, just the name of genes is used, in others - with the addition of Latin names of plants (lines 84, 217, 225). The hyphen and slash in words must be written together (lines 48, 85, 92, 117, 277, 321, 329, 374, 379, 381, 385, 394).
line 68 – not peas but pea followed by a Latin name
line 117 – “literature” needs to be replaced by literature
line 245 - b-conglycinin needs to be replaced by β- conglycinin
References: some articles are indicated without doi and with capital letters in the title of the article.
Author Response
Sir,
According to your recommendation I have corrected my manuscript. I have prepared a full diagram which explain my manuscript, I think it will not make you bored now.
Thanks for you valuable recommendation it helped me a lot to improve my manuscript.
Round 2
Reviewer 1 Report
The authors respond all my questions and issues. They modyfied the manuscript accordingly. I do not see any more issues which speak against of publish this paper.
Author Response
Thank you Sir for your support.
Reviewer 2 Report
Dear Authors,
Thank you for adding a summary diagram that helps read the text, not just makes « it will not make you bored now». But I still have comments.
The abstract is poorly written. In my opinion, it is not necessary to write in the abstract about a small number of works devoted to this problem, but to describe what you have reviewed in the proposed manuscript. In the abstract, it is necessary not only to list what is considered in the review, but also to give some conclusions made in the process of writing the review.
The manuscript does not mention anywhere the pea storage protein vicilin, for which allergic reactions (Sanchez-Monge, R., Lopez-Torrejon, G., Pascual, C. Y., Varela, J., Martin-Esteban, M., & Salcedo, G. (2004). Vicilin and convicilin are potential major allergens from pea. Clinical Experimental Allergy, 34(11), 1747–1753. doi:10.1111/j.1365-2222.2004.02085.x) and amyloid formation (Antonets KS, Belousov MV, Sulatskaya AI, Belousova ME, Kosolapova AO, Sulatsky MI, et al. (2020) Accumulation of storage proteins in plant seeds is mediated by amyloid formation. PLoS Biol 18(7): e3000564. https://doi.org/10.1371/journal.pbio.3000564) are described. Moreover, the sequence that forms the amyloid beta barrel contains methionine.
The manuscript mentions glutathione, but does not disclose the role of glutathione in seed development and germination, although reactive oxygen species and antioxidant protection are likely involved in these processes.
The manuscript is written very sloppily. This is a scientific article, so the simple names of plants, as is customary in a recipe book, must be replaced with Latin names (line 99, 103, 109, 212, 229, 316, 327, 329, 338, 345, 350, 396, etc.). It is necessary to standardize the spelling of amino acids, the manuscript is a complete mess: abbreviated and full, or first enter the full names with abbreviations in brackets, and then use the abbreviations. Arabidopsis without a specific Latin name is not written in italics (line 152, 233, 234, 241, 245, 255, 340, etc.). Mutants (line 233, 288, 290, 360, 362, 364, 370, 372), Latin names (line 292) and gene names (line 340) must be written in Italics. The following abbreviations are not deciphered in the text: MYB28, MYC2, ABA, SMARC1N-PN1, MAPKs (lines 234, 255, 258, 350, 417). In the figure, it is also necessary, regardless of the text, to decipher the following abbreviations: OAS, MYB28, MYC2, ABA, SDI. When using Latin names, the full name is written for the first time (line 285), then the generic name is abbreviated (lines 83, 132, 219, 224, 280). The hyphen in words is written together (lines 278, 332).
Minor remarks:
line 5 - number 1 should be small;
line 12 "sulfur" should not be bold.
The slash in words must be written together (lines 39, 43, 48, 86, 93, 118, 124, 198). Not “and/ or”, “and-/ or” or “NO3- - and/ or SO42-“ but “and/or” or “NO3- and/or SO42-“;
line 364 – what is this “8th DAS”, it’s necessary to decipher;
line 404 – not “metabolism-/ content” but “metabolism/content”;
line 406 – not “phase-/ seed development” but “phase/seed development”;
line 410 – not “leaves and or stem” but “leaves and/or stem”;
line 419 – not “metabolism/ mechanism” but “metabolism/mechanism”;
line 421 – not “nations and or benefited” but “nations and/or benefited”.
The “References” is not corrected according to the rules for authors.
Author Response
Thank you Sir for your support and your valuable recommendations, it will help me to rectify my manuscript. I have included all of your recommendations.
Thanks and regards
Round 3
Reviewer 2 Report
Dear Authors,
I have no complaints about the content of the manuscript, but the design still leaves much to be desired.
Line 22 – not “S-metabolism/ mechanism” but “S-metabolism/mechanism”
Line 47 - you introduced the concept of glutathione, you must give the abbreviation in brackets, since further you use the abbreviation
Line 48 - full name (ROS) is required
Line 49 - you introduced the concept of glucosinolates, you must give the abbreviation in brackets (GSLs), since further you use the abbreviation
Line 51 – full name (Cys, Met) is required
Line 53 – full name (GSSG) is required
Line 54 – full name (GR) is required
Line 54 – use “GSH” instead “glutathione”
Line 56 – use “GSH” instead “glutathione”
Line 77 – use “GSH” instead “glutathione”
Line 88 – use (C. arietinum)
Line 99 – add (Glycine max (L.) Merr.)
Line 104 – use (Brassica napus L.)
Line 104 – use (G. max)
Line 108 – are “canola” and “oilseed rape” the same thing? Please, use one of them
Line 108 – use (B. napus)
Lines 112-113 – use SSP
Line 114 – use (B. napus)
Line 134 – use (Lupinus albus L.)
Line 137 – use M. truncatula, this is not the first mention
Line 159 – you may use Arabidopsis without species name and without italics
Line 159 – use (Oryza sativa L.)
Line 160 – is (SULTR1 to 4) the gene family? If so, use italics
Line 185 – use italics for mutant “sultr4;1/sultr4;2”
Line 212 – do not use italics (Arabidopsis)
Lines 224, 240, 242, 249, 253, 263 – use Arabidopsis without species name and without italics
Lines 232, 236 – use (B. napus)
Line 253 – use (G. max)
Line 263 – use “GSLs” instead “glucosinolates”
Line 272, 278, 282 – use italics for “SDI”
Line 294 – use “B. oleracea”, this is not the first mention of Brassica
Line 301 – use italics for “B. oleracea”
Line 308 – use “Cys and Met” instead “cysteine and methionine”
Line 310 – add (Zea mays L.)
Line 310 – add (Lys)
Line 311 – use “Met” instead “methionine”
Line 312 – use “Lys” instead “lysine”
Line 314 – add (Z. mays)
Line 315 – use “Met” instead “methionine” and “Cys” instead “cysteine”
Line 316, 318, 319, 326, 352, 355, 357 – use “Met” instead “methionine”
Line 317, 318, 348 – use “Cys” instead “cysteine”
Line 336 – use (G. max)
Line 337 – use (P. sativum)
Line 348 – use (G. max)
Line 350 – use “Arabidopsis” instead “Arabidopsis thaliana”
Line 350 – use “cystathionine γ-synthase” instead “cystathionine γ- synthase”
Line 355 – use (G. max)
Line 356 – use (V. ungularis)
Line 361 – use (P. vulgaris)
Line 363 – add (P. sativum)
Lines 364, 366, 367, 372б 373б 374б 375 – it is not necessary to use a capital letter in the name of the vicilin protein
Line 371 – use “P. sativum”
Line 385 – please decipher “ABA”
Line 391 – use italics for mutant “SULTR3”
Line 394 – only “Cys”
Line 400 – use “DAS”
Lines 379-402 – what plant are we talking about? Is it Arabidopsis?
Line 423 – use “B. rapa”
Please correct the capital letters in the titles of the articles in the list of references under the numbers: 7, 11, 39, 86
Author Response
Sir, I have corrected according to your recommendations.
Thanks and regards